# Effectiveness, Tolerability and Prescribing Choice of Antiviral Molecules Molnupiravir, Remdesivir and Nirmatrelvir/r: A Real-World Comparison in the First Ten Months of Use

**DOI:** 10.3390/v15041025

**Published:** 2023-04-21

**Authors:** Cosmo Del Borgo, Silvia Garattini, Carolina Bortignon, Anna Carraro, Daniela Di Trento, Andrea Gasperin, Alessandra Grimaldi, Sara Giovanna De Maria, Sara Corazza, Tiziana Tieghi, Valeria Belvisi, Blerta Kertusha, Margherita De Masi, Ombretta D’Onofrio, Gabriele Bagaglini, Gabriella Bonanni, Paola Zuccalà, Paolo Fabietti, Eeva Tortellini, Mariasilvia Guardiani, Alessandra Spagnoli, Raffaella Marocco, Danilo Alunni Fegatelli, Miriam Lichtner

**Affiliations:** 1Infectious Diseases Unit, Santa Maria (SM) Goretti Hospital, Sapienza University of Rome, 04100 Latina, Italy; 2Department of Public Health and Infectious Diseases, Sapienza University of Rome, 00185 Rome, Italy; 3Department of Neurosciences, Mental Health, and Sense Organs, NESMOS, Sapienza University of Rome, 00189 Rome, Italy

**Keywords:** antivirals, molnupiravir, remdesivir, nirmaltrelvir/ritonavir, early therapy, COVID-19, SARS-CoV2

## Abstract

In 2022, three antiviral drugs—molnupiravir, remdesivir and nirmatrelvir/ritonavir—were introduced for treatment of mild-to-moderate COVID-19 in high-risk patients. The aim of this study is the evaluation of their effectiveness and tolerability in a real-life setting. A single-center observational study was set up, with the involvement of 1118 patients, with complete follow-up data, treated between the 5th of January and the 3rd of October 2022 at Santa Maria Goretti’s hospital in Latina, Central Italy. A univariable and a multivariable analysis were performed on clinical and demographic data and composite outcome, the persistence of symptoms at 30 days and time to negativization, respectively. The three antivirals showed a similar effectiveness in containing the progression of the infection to severe COVID-19 and a good tolerability in the absence of serious adverse effects. Persistence of symptoms after 30 days was more common in females than males and less common in patients treated with molnupiravir and nirmatrelvir/r. The availability of different antiviral molecules is a strong tool and, if correctly prescribed, they can have a significant role in changing the natural history of infection for frail persons, in which vaccination could be not sufficient for the prevention of severe COVID-19.

## 1. Introduction

In the early history of the SARS-CoV-2 pandemic, limited therapeutic possibilities were available. However, thanks to the strong global effort of the international scientific community, some new strategies were developed. In particular, at the very beginning, molecules already employed for treating other diseases, such as hydroxychloroquine, chloroquine, lopinavir–ritonavir and ivermectin [1], but also bioactive natural products and small-molecule inhibitors, were used in order to treat patients with severe COVID-19 [2,3].

Since January 2022, three antiviral drugs against SARS-CoV-2 were introduced in Italy, molnupiravir (MP), remdesivir (RDV) and nirmaltrelvir/ritonavir (NMV/r), available for patients with SARS-CoV-2 infection with a high risk of progression to severe illness. Specifically, MP was introduced on the 29 December 2021 (Agenzia Italiana del Farmaco, AIFA note n°1644/2021); RDV was introduced as a three-day scheme for non-hospitalized patients on the 7 January 2022 (AIFA note n°92/2020); and NMV/r was introduced on 1 February 2022 (AIFA note n°15/2022) [4].

These three antivirals act with different mechanisms of action. In particular, RDV and MP are both prodrugs and act as viral RNA-dependent RNA polymerase (RdRp) inhibitors, interfering with the genomic replication, but with different mechanisms.

In fact, MP is converted into an active nucleoside b-D-N4-hydroxycytidine by esterases present in the plasma [5], inhibiting SARS-CoV-2 replication by a mechanism known as “lethal mutagenesis” and preventing viral propagation by fostering error accumulation in a process referred to as “error catastrophe” [cit], while RDV (GS-5734) is a phosphoramidite prodrug of an adenosine monophosphate analog, metabolized into its pharmacologic analog adenosine triphosphate, acting as a non-obligate chain terminator [6].

NMV/r is a viral protease inhibitor that binds the enzymatic catalytic cysteine residue (Cys145), blocking the viral assembly; ritonavir acts as a pharmacologic booster of nirmatrelvir, inhibiting the CYP3A4 enzyme in order to maintain high plasmatic levels of nirmatrelvir itself [7].

Registration studies, such as “MoveOut” for MP, “Pinetree” for RDV and “Epic-Hr” for NMV/r, have shown similar and good tolerability for the three drugs, but different results in terms of efficacy defined as no progression to death or hospitalization [8,9,10]. In fact, RDV and NMV/r were associated with greater efficacy compared to placebo, reporting a relative reduction in death and hospitalization by 89% and 88%, respectively. MP reported a relative reduction in death and hospitalization by 30%.

However, these studies presented some limits. First of all, people with mild-to-moderate SARS-CoV-2 infection were enrolled during the prevalence of the Delta variant of SARS-CoV-2, but in clinical practice, the antiviral drugs were used during the prevalence of the Omicron variant [8,9,10]. Furthermore, among the people who received antiviral treatment, a high portion did not match the clinical characteristics required to be a candidate for early treatment. In particular, people enrolled had a median age of 40, had no more than one comorbidity, were not vaccinated, and patients affected by immunodeficiency were not considered in these studies.

Considering these limitations, along with the low number of studies that compare the three drugs [11,12,13], this study has the aim of comparing the three molecules in a real-life setting, evaluating their impact in an everyday clinical practice. In other terms, this study has the purpose of evaluating and comparing the three drugs in terms of effectiveness and tolerability, in order to understand which of them could be more suitable for the peculiar clinical features of each patient and offer a personalized treatment.

## 2. Materials and Methods

### 2.1. Study Design

This is a single-center observational real-world study (RWS) of a cohort of patients with a confirmed diagnosis of SARS-CoV-2, through a positive nasopharyngeal swab (NPS). Only non-hospitalized patients with mild-to-moderate COVID-19 disease and one or more risk factors for progression to severe illness, as defined by the European Medicines Agency (EMA) [14] and AIFA guidelines [4], were considered eligible for early treatment. According to the aforementioned guidelines, risk factors include: body mass index (BMI) >30, diabetes mellitus (DM), chronic kidney failure (CKD), immunodeficiency, neurological diseases, cardiovascular diseases, lung diseases, age >65, hospitalization for other diseases, chronic hepatopathy, active oncological diseases and haemoglobinopathies.

Patients treated with early RDV (three-day scheme) who were hospitalized for other diseases different from COVID-19 illness or were in the emergency room were excluded from the study in order to make the three groups of treatment more homogeneous, including only outpatients. In fact, inpatients showed a different baseline from outpatients in terms of severity of comorbidities and clinical symptoms not directly related to SARS-CoV-2 infection, which had an impact on the progression of COVID-19 disease.

### 2.2. Study Setting

A clinic for early COVID-19 was set up at Santa Maria Goretti Hospital, in Latina, Central Italy, at the beginning of March 2021, and was dedicated to providing early treatment for COVID-19 to high-risk outpatients. From the 5 January, when all three antivirals were available, to the 3 October 2022, 3206 patients were treated, when VOCs (variants of concern) Omicron BA.1, BA.2, BA.4 and BA.5 were prevalent in Italy [15].

Patients were considered suitable for therapy if they tested positive for SARS-CoV-2 (NPS) and had at least one risk factor for severe COVID-19, as indicated by AIFA [4]. There were three main possible ways of recruitment: referral by general practitioner (GP), hospital specialists, or self-referral by a phone regional system, as shown in Figure 1. After receiving the application, patients suitable for the enrollment were reached by telephone for an opening counseling considering their clinical features, such as weight and height, to calculate filtrate glomerular rate (FGR), and social conditions. Firstly, general clinical conditions and COVID-19-related symptoms were investigated, in order to stratify risk, then polypharmacy was evaluated, especially potential interactions with NMV/r since it cannot be prescribed if patients’ home therapy includes drugs such as new oral anticoagulants (e.g., Apixaban), atypical antipsychotic drugs (e.g., Quetiapine) and other drugs [16]. Oral therapy with MP or NMV/r was also not considered if patients were dysphagic, preferring intravenous therapy with RDV, in absence of FGR lower than 30 mL/min or alanine aminotransferase (ALT) five times higher than normal levels. When the choice of hospital intravenous administration of RDV was made, and if the patient could not come to the clinic on their own, the hospital provided ambulance transportation.

Only after acquiring patients’ consensus was the adequate molecule prescribed with the right posology. If an oral antiviral was chosen, it was collected by a patient’s relative. Meanwhile, patients treated with RDV were evaluated at the clinic by a medical and nurse team and its 2 h administration was done while monitoring vital signs. In the following days, higher risk patients were monitored by a telemedicine system and the ones who showed any signs of worsening were invited to the emergency department and admitted when needed.

After 30 days from the start of therapy, a telephone follow-up was performed and clinical data about the effect of the three molecules were collected. In particular, we evaluated the persistence of symptoms (e.g., dyspnea, arthromyalgia, fever, cough, rhinitis, gastrointestinal problems, asthenia), evolution of illness (pneumonia, acute respiratory distress syndrome (ARDS), hospitalization or death), time to negativization and eventual adverse effects. In more details we considered ARDS as established by the Berlin definition [17], based on a pO_2_/FiO_2_ ratio <300.

Patients also received a diary in which they could annotate, for 30 days, the presence of COVID-19-related symptoms, adverse effects and vital signs. Patients treated with parenteral RDV reported the presence of side effects through interviews; in other cases, clinicians who were present during the administration of the drug directly observed early adverse reactions. In a minority of cases, if telephonic follow-up could not be performed, clinical data were collected by consulting a regional platform of COVID-19 positivity or medical records of hospitalization.

The EMA and AIFA’s guidelines for excluding patients from one treatment rather than another were strictly followed.

### 2.3. Patient Characteristics

The demographic data collected were age and sex; clinical data were SARS-CoV-2 vaccinal status (date of the last dose), comorbidities, home therapy, necessity of transport to hospital, persistence of COVID-19-related symptoms after 30 days, progression to severe illness (pneumonia, ARDS) or death (COVID-19 and no COVID-19), time to negativization and adverse effects.

For comorbidities, a focus on immunocompromised patients was conducted, and they were further subdivided into groups based both on their disease or therapy: hematological, solid tumors, HIV infection, transplant patients, autoimmune diseases which require chronic immunosuppressive therapy (rheumatological and neurological ones, prevalently) and any other immunosuppressant comorbidities (e.g., diabetes mellitus).

### 2.4. Outcome

The aim of this study is to compare the three antiviral drugs in terms of effectiveness, tolerability and prescribing choice, considering these endpoints: composite endpoint (pneumonia, ARDS, COVID-19 and non-COVID-19-related death) in all the patients and in the immunocompromised subgroup;persistence of symptoms at 30 days (assessed by phone call);NPS negativization (according to the date reported in the regional platform of COVID-19 patients).

We also evaluated the percentage of the most common adverse effects for the three molecules, such as diarrhea, fever, nausea and vomiting, post-infusion tachycardia, hypertension, rash, headache, mucositis, hypotension, dizziness, metallic taste, inappetence, increased liver markers, abdominal pain and fatigue.

To study the persistence of symptoms at 30 days, according to the criteria established by the National Institute for Health and Care Excellence (NICE) and the World Health Organization (WHO), we studied which demographic and clinical features could influence the presence of post–acute COVID-19 syndrome, defined as a set of signs and symptoms that emerge during or after an infection consistent with COVID-19, persist for more than 12 weeks and are not explained by an alternative diagnosis [18,19]. More precisely, many experts, including the NICE panel, agree with subdividing into two categories the immediate outcome: a post COVID-19 subacute phase of ongoing symptoms that lasts from 4 to 12 weeks after the onset of illness, and a chronic-phase or long COVID-19, defined as symptoms and abnormalities that last more than 12 weeks after the onset of illness and are not explained by other causes [18,19].

### 2.5. Statistical Analysis

Data are expressed as mean and standard deviation (SD) or median and interquartile range (IQR) for numerical variables according to their distribution, and as counts and percentages for categorical variables. Comparison between treatment groups was performed using a chi squared test or Kruskal–Wallis test, as appropriate. Multivariable logistic regression models were estimated to establish the influence of covariates (age, sex, treatment, immunodeficiency, neurological diseases, chronic kidney disease, liver dysfunction, vaccination) on the outcomes (composite endpoint, persistence of symptoms at 30 days). A multivariable linear regression model was defined to evaluate the impact of the covariates on time until negativization. Model selection was performed using a stepwise procedure based on the Akaike Information Criterion.

*P* values < 0.05 were considered to be significant. Confidence intervals were at the 95% level. All analyses were performed using R software (version 4.2.2, R Foundation for Statistical Computing, Vienna, Austria).

## 3. Results

From 5 January to 3 October 2022, a total of 1118 patients were treated, 230 were treated with RDV, 499 with MP and 389 with NMV/r, as shown in Table 1.

### 3.1. Demographic and Clinical Data of the Three Populations

Age analysis shows that MP was prescribed more often in older patients in comparison to the other two groups of treatment, as shown in Table 1.

Considering the variable sex, no statistically significant difference was observed between the three groups of patients.

Among risk factors, it was observed that in patients affected by immunodeficiency, RDV was preferred, while in patients with neurological and cardiovascular diseases and CKD, MP was preferred.

Among patients with altered immunological status, those who suffered from hematologic disease were mostly treated with RDV or NMV/r, with a statistically significant difference versus MP (*p* = 0.016), as shown in Table 2.

Patients who received organ transplant were mostly treated with RDV, with a statistically significant difference between MP and NMV/r (*p* = 0.001).

Finally, MP was associated more frequently with patients affected by cardiovascular disease and diabetes mellitus (Other in Table 2), confirming the descriptive clinical data analysis about comorbidities of all treated patients.

### 3.2. Analysis of End Points

A multivariate analysis was performed with a multivariable logistic regression model to analyze the differences between the three groups of treatment.

The primary endpoint was the clinical progression, defined as the above composite outcome in all patients treated and in the immunocompromised subgroup. Secondary endpoints were the persistence of symptoms at 30 days and the negativization period.

Regarding clinical evolution, progression to pneumonia, ARDS, COVID-19 or non-COVID-19-related death appears to be very low. This was similar for all three drugs (progression was observed in the 2.8% of patients treated with MP, 1.3% of those treated with NMV/r and 3% of patients treated with RDV). There were four documented COVID-19-related deaths: three in the MP-treated group and one in the RDV-treated group (Table 3).

A statistically significant difference in terms of time to negativization was observed (Table 3). In particular, a shorter time was observed in the NMV/r group (median 8 days, IQR 7–10) compared with the other two molecules.

From the univariate analysis among the immunocompromised subgroup, no statistically significant difference was found between the three groups of treatment in terms of clinical progression of SARS-CoV-2 infection to severe patterns of disease and in terms of all-cause mortality (COVID-19 and non-COVID-19), as shown in Table 4.

A difference statistically significant in time to negativization emerged between the NMV/r’s group (*p* < 0.001) of treatment and the other two groups. In fact, as Table 4 below shows, NMV/r seems to be related to early negativization of NPS in immunocompromised patients as well (median days 8, IQR 7–10 in NMV/r vs. median days 10, IQR 9-13 both in RDV and MP).

Age, comorbidities such as immunodeficiency (OR = 6.14; IC = 2.29, 17.20), CKD (OR = 7.98, IC = 1.56, 14.26) and neurological issues (OR = 4.65; IC = 1.48, 13.38) seem to be related to a high risk of progression of COVID-19 illness. Complete vaccination appears to be a protective factor (OR = 0.22; IC = 0.06, 1.07) (Figure 2).

Furthermore, patients treated with MP and NMV/r showed a significantly lower persistence of symptoms at 30 days compared to the group treated with RDV, as the univariate analysis pointed out (MP vs. RDV OR = 0.46; IC = 0.30, 0.71, NMV/r vs. RDV OR = 0.56; IC = 0.37, 0.85) (Figure 3). Additionally, in general, females (OR = 1.68; IC = 1.20, 2.37) and patients who suffer from pulmonary diseases seem to be more affected by long-term symptoms (OR = 1.65; IC = 1.13, 2.39).

As illustrated in Figure 4, time to negativization seems to be shorter in patients treated with NMV/r than in patients who received MP or NMV/r as medications. This is confirmed by the multivariable analysis, as shown in Figure 4; not only NMV/r (beta = −1.84; IC = −2.70, −0.98) but also vaccination (beta = −1,93; IC = −3.13, −0.74) seem to be a protective factor for shorter time to negativization. On the other hand, age (beta = 0.02; IC = 0.00, 0.04) seems to be a risk factor for longer time to negativization.

### 3.3. Adverse Effects

Although no severe adverse effects, according to the EMA definition [20], were reported in the three groups of treatment, RDV showed the fewest number of events (14.8%); MP and NMV/r, on the other hand, showed a number of events in 22.5% and 54% of cases, respectively (Figure 5), mainly diarrhea and metallic taste (Figure 6). Only 13 patients voluntarily interrupted early treatment with antiviral drugs: five patients treated with MP, for diarrhea and urticarial rash onset, six with NMV/r, complaining of nausea and vomiting, and two with RDV. However, it must be pointed out that these latter were not for the onset of adverse effects but rather because one patient decided on his own to not continue the treatment and the other one was converted to a five-day scheme therapy with RDV after a thorax CT scan documented COVID-19-related bilateral interstitial pneumonia.

## 4. Discussion

This study suggests that the heterogeneity of antiviral drugs available for the prevention of severe SARS-CoV-2 infection in high-risk patients plays a key role in terms of patient management, offering the possibility to choose the most suitable drug for every single patient while ensuring similar clinical outcomes and significant containment of disease progression.

Considering the results of this study, elderly patients preferred to be treated with MP, instead of NMV/r or RDV, probably because of conspicuous drug interactions with NMV/r that impede the prescription of this molecule in this age range. Furthermore, MP’s prescription is more consistent in patients with comorbidities such as cardiovascular diseases, neurological diseases and CKD that require chronic therapy with drugs that cannot be safely associated with NMV/r [16] nor modified in their posology. On the other hand, CKD could be associated with high levels of creatinine and low FGR that excludes the possible prescription of RDV or NMV/r without any risk. Similar results were obtained through a multicenter observational study—the FEDERATE cohort—confirming the importance of the availability of MP as an alternative drug that could be prescribed in high-risk patients, including in the early treatment of COVID-19 in people who could otherwise be excluded because of important contraindications to NMV/r or RDV [13].

The cohort of patients involved in this study was mostly vaccinated with a complete vaccination cycle; the RDV group contained the largest group of patients with an incomplete vaccination cycle (13.9%). Some hypotheses can be made in order to explain this result: patients who refused vaccination or did not complete it may be more confident with a therapy with a longer post-marketing period at the time of the study, with a direct 3 h medical supervision during treatment and also a shorter time of administration. However, it must be pointed out that we did not perform any questionnaires to evaluate patients’ preferences, and so this remains a mere supposition.

Among the three antiviral drugs, there were no statistically significant differences between the three groups of treatment, concerning both evolution to pneumonia/ARDS and death, as other research has established [11,12,13]. Risk analysis in our study underlines that immunodeficiency and liver disease presented the higher risk of progression (OR 6.14 and 7.98; IC 95%, respectively), followed by CKD and neurological diseases (OR 4.92 and 4.65, respectively), confirming evidence deriving from several other studies [10]. Immunodeficiency remains a challenge for the COVID-19 pandemic, especially in the Omicron era; in fact, VOCs Omicron, despite their lower pathogenic role compared to other strains (e.g., Delta), are more transmittable and have high power of immune-escape, even from vaccine-induced immunity, which remains the first line of prevention of severe disease in immunodeficient patients [21]. Other studies also underlined the aggressivity of Omicron variants, associated with an increased risk of severe clinical patterns in immunocompromised patients [13,22]. In this setting, some scientists have proposed a new approach with new antivirals, better association of two antivirals, or a combination of antiviral and monoclonal antibodies (mAbs) [23,24,25]. Among monoclonal antibodies, tixagevimab–cilgavimab (Evusheld), which has shown efficacy in prophylaxis of SARS-CoV-2 infection in immunocompromised patients both in clinical trials [26] and RWS [27], also has a possible therapeutic role if administered alone or in association with antiviral molecules to immunocompromised patients infected by SARS-CoV-2 [28]. Unfortunately, in vitro studies demonstrated a significant and many-fold increase of the minimal inhibitory concentration against more recent omicron variants for all mAbs available [29,30]. However, emerging data showed a significant clinical impact of these mAbs, and other studies are necessary in order to understand their role in the future and their possible synergy with antiviral drugs.

Other studies have assessed the necessity of a “tailored and standardized” therapeutic approach in the cases of immunocompromised in- and outpatients with SARS-CoV-2 infection [31], with a particular attention on patients with B-cell depletion due not only to primary severe immunodeficiency but also to biological therapy with Rituximab (anti-CD20 mAb) or in treatment with Fingolimod; in fact, if not properly detected and treated these patients have a COVID-19 case fatality rate of 40% [31,32].

Some good evidence was obtained from this study. In fact, in the immunocompromised subgroup (treated with a monotherapy regimen—only one of the three antivirals, not in combination with mAbs), the three antiviral molecules seem to have the same behavior from a clinical and therapeutic point of view that they show in immunocompetent patients, affected by other comorbidities not referable to an altered immunological status (Table 4). This result suggests the importance of the availability of antiviral molecules as a powerful therapeutic presidium that could be strengthened if associated with proper and effective mAbs.

Concerning the second endpoint, patients affected by lung diseases (OR = 1.65; IC = 1.13, 2.39) seem to be associated with a major risk of persistence of symptoms at 30 days, probably because of the chronic lung dysfunction that could impede a rapid recovery from respiratory COVID19-related symptoms. Moreover, male sex seems to be associated with a lower risk of having 30-day symptoms.

Despite that, it is important to keep investing in studies and research about risk factors and comorbidities associated with persistent COVID19-related symptoms, in order to find out the way to manage specific groups of patients during the acute infection to prevent post-COVID-19 syndrome. Previous studies have shown in general that women showing a major persistence of symptoms at 30 days is a common finding. In particular, they underline how women reported symptoms that constrained daily activities more than men [33], and that female patients were more likely to have headaches, myalgia and abdominal symptoms, and less likely to have abnormal breathing and cognitive deficits than male patients [33].

Concerning antivirals, the risk of COVID19-related symptoms at 30 days was significantly decreased for patients treated with MP and NMV/r compared to the group treated with RDV.

Considering then the persistence of symptoms at 30 days and the study of Post COVID-19 syndrome, the University of Oxford and the National Institute for Health and Care Researches have recently set up a clinical study called PANORAMIC. The study aims to find out in which patients the new antivirals acted properly, preventing the need of hospital admission and increased recovery speed. In particular, this study is open to anyone with ongoing COVID-19 symptoms and a positive PCR test [34]. In the near future, a lot of new data will be collected about long COVID-19 and new clinical strategies could be pointed out for the management of some categories of patients.

The endpoint time to negativization seems to be influenced positively by treatment with NMV/r and by vaccination (Figure 4), confirming the strength of anti-SARS- CoV-2 vaccine and its ability to induce a good immunological response with adequate antibody production, which is necessary to put in place as an early efficient weapon against SARS-CoV-2 infection.

Vaccination remains the most powerful presidium against severe COVID-19 illness, despite the availability of antiviral drugs, and it is fundamental to prevent the progression to severe COVID-19 disease and has changed the natural history of SARS-CoV-2 infection. However, another study has affirmed that vaccination did not reduce the risk of anxiety/depression, headache, abdominal symptoms, chest/throat pain, abnormal breathing and cognitive symptoms in patients who suffered from long-COVID, but that certain symptoms, notably fatigue and myalgia, were less common in the vaccinated population [35].

Regarding the same effectiveness of the three antivirals in containing the progression of COVID-19 disease, this study suggests that in the Omicron era, early therapy has a strong impact on the natural history of the infection [36,37] and confirms the importance of real world studies (RWS) as instruments to validate the results of clinical trials. Antivirals play a key role in the Omicron era, especially NMV/r, which showed a potential therapeutic efficacy against this novel variant in previous in vitro studies. Despite the biological mutation of Omicron, both in RNA-dependent RNA-polymerase (RpRd)—the therapeutic target of RDV and MP—and the SARS-CoV-2 major protease inhibitor—the target of NMV/r [18,19]—these drugs maintain therapeutic efficacy, confirmed by several real world studies [13,38,39,40,41].

The absence of severe adverse effects [13] in the three groups of treatment underlines the possibility of a safe prescription that could reassure patients who show reticence to the consumption of the antiviral drugs. In the group treated with MP, the most common adverse effect was diarrhea, limited to the days of consumption of the drug. Other reviews and RWSs have demonstrated that gastrointestinal discomfort (nausea and diarrhea) and headache was mostly reported by patients treated with MP [5]. Dysgeusia, referred to as a metallic taste by patients, was associated in our study mainly to NMV/r and was very frequent and transitory, as in other studies [5,42,43]. The mechanism for this adverse effect is not clear but a study pointed out that ritonavir, and protease inhibitors in general, could have a role in modifying the taste perception of a variety of taste compounds, influencing patients’ compliance with medical treatment regimens [44]. The biological mechanisms involved in this phenomenon seem to be related to adverse sensory properties of the drug itself and to biochemical disruption of normal taste and smell signals caused by medications. In general, geriatric patients complain of dysgeusia more than younger ones, possibly because of polypharmacy [45]. In our study we did not perform blood test analysis because of the real-life nature of the study in a cohort of outpatients. A gastroenterologic study investigated the possible liver toxicity of MP and NMV/r, affirming that there is a minimal risk of drug-induced liver injury (DILI); in fact, compared to no antiviral treatment, both MP and NMV/r did not increase the risk of elevated liver enzymes or DILI [46].

A final mention must be made of the recent discontinuation of MP as established by AIFA on the 15th of March (AIFA DG/85/2023) [47], after the EMA recommended the refusal of its marketing authorization [48]. In fact, the Agency’s opinion was that it was not possible to conclude that MP could reduce the risk of any outcomes in adults at risk of severe disease and that its balance of benefits and risks in the treatment of COVID-19 could not be established. However, we must stress that this is not what we observed in our RWE study, and also in other studies [49], and that we confirmed the absence of safety issues related to MP as remarked on both by EMA and AIFA.

## 5. Limitation of the Study

The main limitation of this study is the retrospective nature of the study and the absence of an untreated control group.

Furthermore the data collection of our study is mostly based on phone call follow-up, so the evaluation of symptoms at 30 days are self-reported and they were not objectified by physical and instrumental examination or laboratory tests. Although patients who reported important persistent symptoms were invited for a medical consultation at our post-COVID-19 clinic.

Finally, it must be considered that no specific objectification with NPS for time to negativization was provided by our clinic, in fact it was self-referred from the patients themselves and it has to be considered as an estimate and, whenever possible, it was double-checked by verifying the date reported in the local regional platform.

## 6. Conclusions

In conclusion, our real-world evidence study compared antiviral treatments for early COVID-19 in a high risk population demonstrating a low rate of progression in all treatment groups without differences. Each patient received the best drug considering comorbidities, personal choice and polypharmacy, and side effects were limited and discontinuation was rare. A strong network and a rapid communication between GPs and hospital teams remains essential, in order to detect high risk patients early and reach the maximum effectiveness of treatment. The best option for the severe immunocompromised population remains to be addressed in future studies.

## Figures and Tables

**Figure 1 viruses-15-01025-f001:**
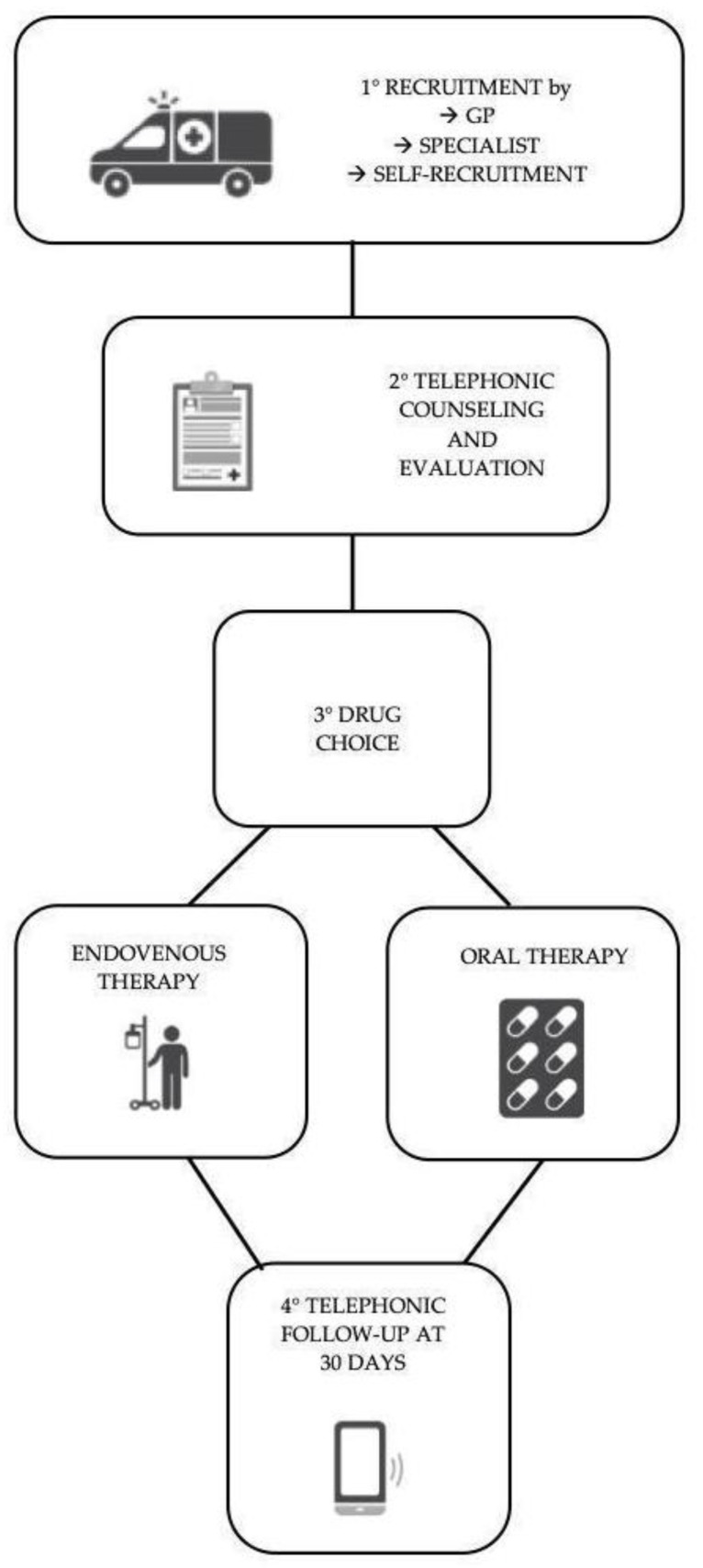
Patients’ recruitment algorithm.

**Figure 2 viruses-15-01025-f002:**
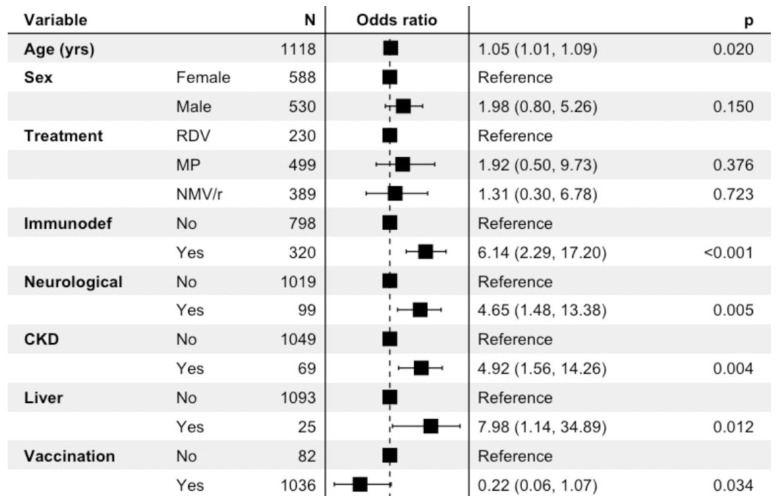
Logistic regression multivariate analysis of composite outcome (pneumonia, ARDS, COVID-19 Death, Non-COVID-19 Death) YRS: years; RDV: remdesivir; MP: molnupiravir; NMV/r: nirmatrelvir/ritonavir; CKD: chronic kidney disease.

**Figure 3 viruses-15-01025-f003:**
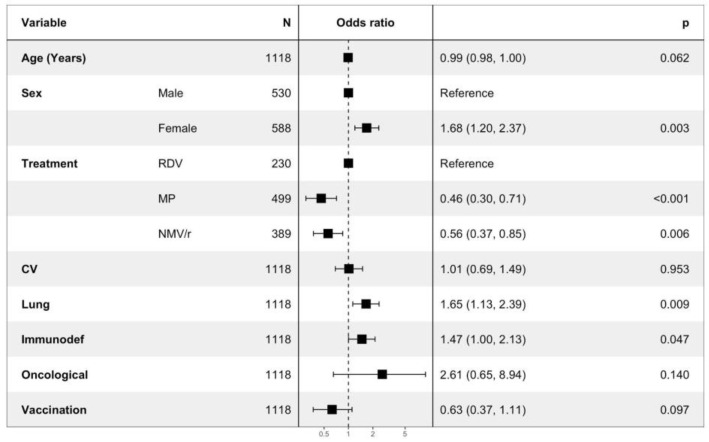
Logistic regression analysis of persistence of symptoms at 30 days. RDV: remdesivir; MP: molnupiravir; NMV/r: nirmatrelvir/ritonavir; CV: cardiovascular disease; Immunodef: immunodeficiency.

**Figure 4 viruses-15-01025-f004:**
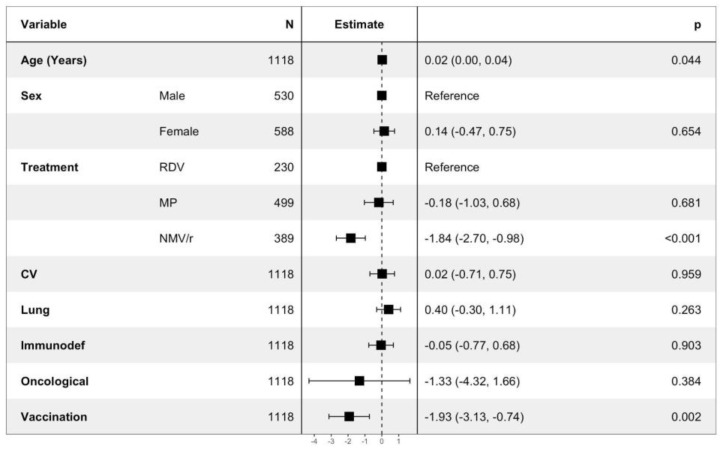
Linear regression analysis of time to negativization. RDV: remdesivir; MP: molnupiravir; NMV/r: nirmatrelvir/ritonavir; CV: cardiovascular disease; Immunodef: immunodeficiency.

**Figure 5 viruses-15-01025-f005:**
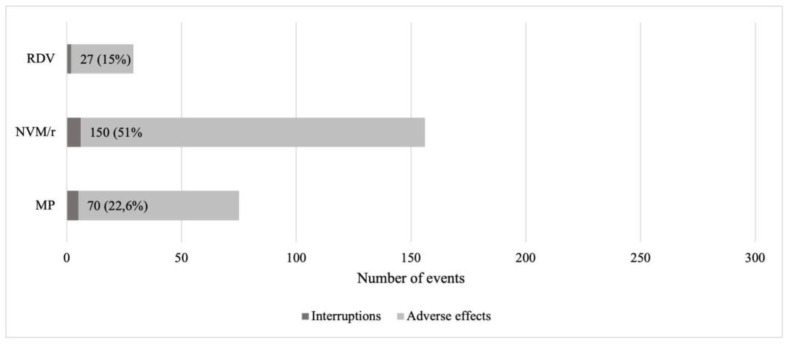
Frequency of self-reported adverse effects and interruptions in the three groups of treatment. MP: molnupiravir, NVM/r: nirmatrelvir/ritonavir, RDV: remdesivir. MP vs. RDV *p*-Value 0.0001; NVM/r vs. RDV *p*-Value 0.0001.

**Figure 6 viruses-15-01025-f006:**
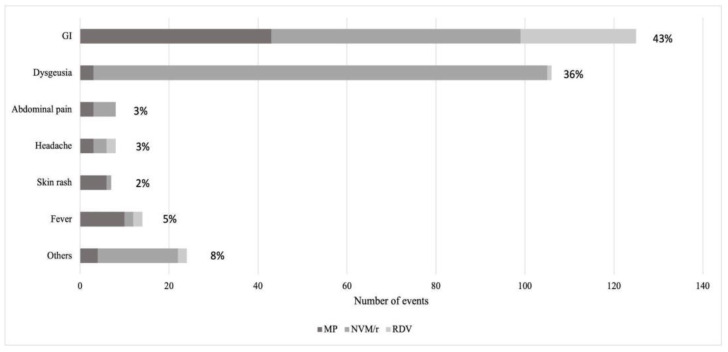
Adverse effects reported by patients GI: gastrointestinal. MP: molnupiravir, NMV/r: nirmatrelvir/r, RDV: remdesivir.

**Table 1 viruses-15-01025-t001:** Univariable analysis of demographic and clinical data of patients. MP: molnupiravir; RDV: remdesivir; NMV/r: nirmatrelvir/ritonavir; CKD: chronic kidney disease.

Total Patients 1118	RDV *n* = 230	MP *n* = 499	NMV/r *n* = 398	*p*-Value
Age median (min, max)	66 (18, 98)	78 (21, 103)	64 (17, 104)	<0.001
Sex n (%)	116 (50.4)	247 (49.5)	167 (42.9)	0.089
Incomplete vaccinal status n (%)	32 (13.9)	26 (5.2)	24 (6.2)	<0.001
Immunodeficiency n (%)	94 (40.9)	97 (19.4)	129 (33.2)	<0.001
Cardiovascular disease n (%)	130 (56.5)	367 (73.5)	175 (45)	<0.001
Neurological disease n (%)	9 (3.9)	67 (13.4)	23 (5.9)	<0.001
CKD n (%)	78 (3.5)	48 (9.6)	13 (3.3)	<0.001

**Table 2 viruses-15-01025-t002:** Univariable analysis of immunodeficient patients’ subgroup. Others: diabetes mellitus, cardiovascular disease, Down’s syndrome and other rare diseases (Proteus syndrome, Shwachman–Diamond syndrome).

Total Patients 320	RDV *n* = 94	MP *n* = 97	NMV/r *n* = 129	*p*-Value
Hematologic disease n (%)	20 (21.3)	8 (8.2)	28 (21.7)	0.016
Solid tumor n (%)	30 (31.9)	44 (45.4)	49 (38)	0.160
Organ transplant n (%)	7 (7.4)	1 (1.0)	0 (0.0)	0.001
HIV infection n (%)	5 (5.3)	6 (6.2)	5 (3.9)	0.722
Immunosuppressive therapy n (%)	30 (31.9)	22 (22.7)	36 (27.9)	0.357
Other n (%)	19 (20.2)	32 (33.0)	26 (20.2)	0.048

**Table 3 viruses-15-01025-t003:** Results of univariable analysis of patients’ clinical evolution by the two endpoints: composite outcome and symptoms at 30 days. * Composite outcome: pneumonia, ARDS, COVID-19 Death, Non-COVID-19 Death. IQR: interquartile range.

Totals Patients 1118	RDV *n* = 230	MP *n* = 499	NMV/r *n* = 389	*p*-Value
OUTCOME				
Clinical Progression *	3 (1.3)	14 (2.8)	5 (1.3)	0.194
Time to negativization Median [(IQR)]	10 [9,10,11,12]	10 [8,9,10,11,12,13]	8 [7,8,9,10]	<0.001
All cause mortality (COVID-19 and no COVID-19) n (%)	2 (0.9)	7 (1.4)	4 (1)	0.785
COVID-19 mortality n (%)	1 (0.4)	3 (0.6)	0 (0)	0.261
Symptoms at 30 days n (%)	59 (25.7)	56 (11.2)	62 (15.9)	<0.001

**Table 4 viruses-15-01025-t004:** Results of univariable analysis of immunocompromised subgroup by the endpoints: composite outcome (clinical progression), all-cause mortality (COVID-19 and non-COVID-19) and time to negativization. * Composite outcome: pneumonia, ARDS, COVID-19 Death, Non-COVID-19 Death. IQR: interquartile range.

Totals Patients 320	RDV *n* = 94	MP *n* = 97	NMV/r *n* = 129	*p*-Value
OUTCOME				
Clinical Progression * n (%)	2 (2.1)	5 (5.2)	4 (3.1)	0.499
All–cause mortality (COVID-19 and no COVID-19) n (%)	1 (1.1)	2 (2.1)	4 (3.1)	0.587
Time to negativization Median [(IQR)]	10 [9,10,11,12,13]	10 [9,10,11,12,13]	8 [7,8,9,10]	<0.001

## Data Availability

The raw data supporting the conclusions of this article will be made available by the authors without undue reservation.

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
