# Peer review of "Effectiveness, Tolerability and Prescribing Choice of Antiviral Molecules Molnupiravir, Remdesivir and Nirmatrelvir/r: A Real-World Comparison in the First Ten Months of Use"

_viruses, 2023, doi:10.3390/v15041025_

Round 1

Reviewer 1 Report

The authors present observational single-center data without control arm. This type of data could be released from any center across the world at the time ofa pandemic. Without control arms or cost-efficacy analyses, this type of data has very little value to the prescribers community and very low chances of being cited. Furthermore, they totally forgot to mention that one of the 3 drugs, has been deauthorized by both EMA and AIFA because of INEFFICACY, and there are serious concerns about mutagenicity. That clearly shows that uncontrolled sciences have very low value.

Reviewer 2 Report

Dear authors,

I have read your exciting manuscript about the use of early antiviral treatment for SARS-CoV-2 infections. Few data about this topic are available in the literature, and data like yours are fundamental.

However, some issues are present.

General comment

Please modify Sars-Cov-2 in SARS-CoV-2 and Covid-19 in COVID-19.

Sometimes you confuse SARS-CoV-2 with COVID-19. For example, in the methods section you wrote “confirmed diagnosis of Covid-19, through a positive nasopharyngeal swab”. In this case, you should write “confirmed diagnosis of SARS-CoV-2, through a positive nasopharyngeal swab”. Again you wrote COVID-19 vaccination, but it should be correct to write SARS-CoV-2 vaccination. Please read the manuscript carefully and fix this issue.

Abbreviations should be written entirely in the first appearance in the text. Please, check the paper carefully (e.g. FGR has been written entirely in the second appearance (line 109 instead of 103).

Introduction

Regarding the mechanism of action of MP and RDV, I suggest adding that it is pretty different since MP causes random mutation (https://doi.org/10.1038/s41594-021-00657-8), resulting in the so-called error catastrophe. Instead, RDV is a nonobligate chain terminator.

I also suggest writing in the introduction the result of the Panoramic Trial (https://doi.org/10.1016/S0140-6736(22)02597-1), describing its limitations.

I suggest adding that the main problem of NMV/r is the DDIs.

Methods

Line 97: the definition of NPS has already been explained (line 76).

I suggest explaining what you mean by ARDS. You considered it when P/F is < 300, or 200?

Result

I do not understand why dividing table 1 and table 2. Therefore, I suggest joining them. In addition, I also suggest calculating the p-value for table 1.

I do not understand if the logistic regression in Figure 2 is a multivariate or univariate analysis

Reviewer 3 Report

I recommend this paper to be published in the journal. Here are some suggestions:

1: As the COVID-19, an outbreak caused by the Severe Acute Respiratory Syndrome Coronavirus 2 (SARS-CoV-2), is still spreading and this study may have a certain amount of readership. In general, this clinical experience is sound, but take “Covid-19 infection” (lines 36, 60, 230, 289, 329, 337) for example, the authors should carefully review the manuscript to unify as “SARS-CoV-2 infection”. This concern should be taken into consideration.

2. “Sars- Cov2”, “SARS-CoV2”, “Sars- CoV2”, and “Sars-CoV2”. The authors should carefully review the manuscript to unify as “SARS-CoV-2”.

3. There is a lack of recent literature citations, the authors should enrich the related articles. For example, in page 11 line 325-327, “In this setting some scientists propose a new approach with new antivirals or better association of two antivirals (Biomedicines. 2021, 9, 689) or combination of antiviral and monoclonal antibodies (mAbs) [19].”

4. “Keywords” of the manuscript should more concise.

5. It is suggested to add some background in introduction (in page 1 line 34-36, “Since January 2022, three antiviral drugs …… to severe illness.”) and highlight the novelty of this work clearly. For example, “Traditional medicine (Processes. 2020, 8, 937), bioactive natural products (Front. Pharmacol. 2022, 13, 926507), small-molecule inhibitors (Pharmaceuticals. 2022, 15, 165) playing an irreplaceable role in the treatment of SARS-CoV-2 infection. However, promising drugs do not exist.”

6. RNA- dependent RNA-polymerase (RpRd) ---RNA-dependent RNA-polymerase (RdRp).

Round 2

Reviewer 1 Report

The authors did not respond adequately to any of my concerns

Reviewer 3 Report

Accept in present form